# Synthesis and Validation of TRIFAPYs as a Family of Transfection Agents for Therapeutic Oligonucleotides

**DOI:** 10.3390/biom14040390

**Published:** 2024-03-25

**Authors:** Berta Isanta, Ana Delgado, Carlos J. Ciudad, Mª Antònia Busquets, Rosa Griera, Núria Llor, Véronique Noé

**Affiliations:** 1Department of Pharmacology, Toxicology and Therapeutic Chemistry, School of Pharmacy and Food Sciences, University of Barcelona, 08028 Barcelona, Spain; bertaisanta01@gmail.com (B.I.); griera@ub.edu (R.G.); nllor@ub.edu (N.L.); 2Department of Biochemistry and Physiology, School of Pharmacy and Food Sciences, University of Barcelona, 08028 Barcelona, Spain; anadeelgado@icloud.com; 3Nanoscience and Nanotechnology Institute, IN2UB, University of Barcelona, 08028 Barcelona, Spain; mabusquetsvinas@ub.edu; 4Department of Pharmacy and Pharmaceutical Technology and Physical Chemistry, School of Pharmacy and Food Sciences, University of Barcelona, 08028 Barcelona, Spain

**Keywords:** transfection, therapeutic oligonucleotide, cationic liposome, PPRH, prostate cancer, breast cancer, apoptosis, delivery

## Abstract

Transfection agents play a crucial role in facilitating the uptake of nucleic acids into eukaryotic cells offering potential therapeutic solutions for genetic disorders. However, progress in this field needs the development of improved systems that guarantee efficient transfection. Here, we describe the synthesis of a set of chemical delivery agents (TRIFAPYs) containing alkyl chains of different lengths based on the 1,3,5-tris[(4-alkyloxy-1pyridinio)methyl]benzene tribromide structure. Their delivery properties for therapeutic oligonucleotides were evaluated using PolyPurine Reverse Hoogsteen hairpins (PPRHs) as a silencing tool. The binding of liposomes to PPRHs was evaluated by retardation assays in agarose gels. The complexes had a size of 125 nm as determined by DLS, forming well-defined concentrical vesicles as visualized by Cryo-TEM. The prostate cancer cell line PC-3 was used to study the internalization of the nanoparticles by fluorescence microscopy and flow cytometry. The mechanism of entrance involved in the cellular uptake was mainly by clathrin-mediated endocytosis. Cytotoxicity analyses determined the intrinsic toxicity caused by each TRIFAPY and the effect on cell viability upon transfection of a specific PPRH (HpsPr-C) directed against the antiapoptotic target survivin. TRIFAPYs C12-C18 were selected to expand these studies in the breast cancer cell line SKBR-3 opening the usage of TRIFAPYs for both sexes and, in the hCMEC/D3 cell line, as a model for the blood–brain barrier. The mRNA levels of survivin decreased, while apoptosis levels increased upon the transfection of HpsPr-C with these TRIFAPYs in PC-3 cells. Therefore, TRIFAPYs can be considered novel lipid-based vehicles for the delivery of therapeutic oligonucleotides.

## 1. Introduction

The availability of transfection agents is crucial in the field of gene therapy, as it enables the internalization of nucleic acids for therapeutic purposes within eukaryotic cells to modulate or silence gene expression in diseases such as cancer [1]. However, these methods come with limitations in terms of their effectiveness, efficiency, and impact on cell viability [2,3]. Transfection agents are generally classified into viral and non-viral systems [4]. Viral systems present a limitation in the size of the transgene to be transferred [5] and it tends to be hard to execute these techniques since they require specialized training for personnel. Additionally, these methods present a high risk of unintended mutations [6], such as random DNA insertions or unwanted immunogenic responses. Non-viral methods include various physical and chemical approaches used in gene therapy [7,8,9]. Physical methods such as electroporation, sonoporation, and microinjection are based on cell membrane deformation, therefore, they compromise cell viability and it is complicated to scale them. Finally, the synthesis of chemical compounds offers greater safety compared to viral transduction and has a lower impact on cell viability than physical methods. Moreover, chemical compounds offer greater ease of production and can be readily modified, thus enhancing the potential for targeted delivery to specific tissues [8,10,11]. However, they still face challenges such as variable levels of internalization, lower efficiency, and some toxicity. The Organic Chemistry section of our research group has been focused on the synthesis of liposome-based chemical compounds. Recently validated compounds included 1,3-bis [(4-oleyloxy-1-pyridinio)methyl]benzene dibromide (DOPY) [12] and 1,3,5-tris[(4-oleyloxy-1-pyridinio)methyl]benzene tribromide (TROPY) [13]. Continuing our previous work, the current project involves the synthesis and validation of a family of compounds named *tricationic fatty acid pyridinium* (TRIFAPYs). This collection includes a total of nine cationic compounds containing alkyl chains of different lengths, from C4 through C20, following the 1,3,5-tris[(4-alkyloxypyridinio)methyl]benzene tribromide structure.

Currently, the development of novel gene therapies represents a significant breakthrough in the treatment of genetic diseases, such as certain cardiovascular, hematological [14,15], neurological [15,16], and oncological conditions. The Food and Drug Administration (FDA) [17,18] has approved the use of nucleic acids for therapeutic purposes [19], including small interfering RNAs (siRNA), aptamers [20], and antisense oligonucleotides (ASOs) [17]. Our research group has developed a genetic silencing tool known as PolyPurine Reverse Hoogsteen (PPRH) [21,22]. PPRHs are formed by two sequences of polypurines arranged in an antiparallel orientation, linked by a thymidine loop. The Reverse Hoogsteen bonds established between guanine–guanine and adenine–adenine bases enable the formation of their clamp-like structure. The mechanism of action of PPRHs relies on the complementary and specific binding to one of the polypyrimidine chains in the genomic DNA by the formation of Watson and Crick bonds [23]. The formation of the triplex leads to the displacing of the double-stranded DNA, culminating in transcription inhibition and, consequently, gene silencing [23,24]. PPRHs can be directed to polypyrimidine sequences found in practically all genes in the promoter, intronic or exon regions [25,26]. The ability of PPRHs to modulate the expression of targeted genes has been confirmed through in vitro [21,27] and in vivo studies conducted within our laboratory [28]. In this work, we tested the collection of TRIFAPYs delivering a PPRH against the promoter region of the antiapoptotic survivin gene (BIRC5) into cancer cell lines and a blood–brain barrier model.

## 2. Materials and Methods

### 2.1. Chemistry: General Remarks

Chemical reagents were obtained from Sigma-Aldrich (St. Louis, MO, USA). All solvents were of analytical grade and used directly without any further purification. Evaporation of solvent was accomplished with a rotatory evaporator. Drying of organic extracts during the workup of reactions was performed over anhydrous Na_2_SO_4_. Thin-layer chromatography was carried out on SiO_2_ (silica gel 60 F254), and the spots were located by UV light and a 1% KMnO_4_ solution. Chromatography refers to flash column chromatography and was carried out on SiO_2_ (silica gel 60, 230–400 mesh). NMR spectra were recorded on a Bruker 400 spectrometer [400 MHz (^1^H) and 100.6 MHz (^13^C)], and chemical shifts are reported in *δ* values, in parts per million (ppm) relative to Me_4_Si (0 ppm) or relative to residual chloroform (7.26 ppm, 77.0 ppm) or methanol (3.31 ppm, 49.0 ppm) as an internal standard. Data are reported in the following manner: chemical shift, multiplicity, coupling constant (*J*) in hertz (Hz), and integrated intensity. High-resolution mass spectra (HMRS) were performed by CCiT-UB using an electrospray (ESI) ionization source and a TOF analyzer.

#### 2.1.1. General Procedure A

Synthesis of 4-alkoxypyridines (1b-i): Alcohol (1.0 equiv.) was added dropwise to a stirring suspension of NaH (1.1 equiv., 90%) in dry DMSO at room temperature, and the mixture was stirred at this temperature (1b, 1e-f) or heated at 60 °C (1c-d, 1g-i) for 30 min. Then, crude 4-chloropyridine (1 equiv. freshly liberated using saturated aqueous NaHCO_3_ from its hydrochloride salt) was added and the stirring was continued at room temperature for 24 h (1b, 1e-f), or 48 h (1e, 1f), or at 60 °C for 48 h (1c-d, 1g-i). The reaction mixture was quenched with H_2_O, and the resulting solution was extracted with EtOAc. The combined organic extracts were dried, filtered, and concentrated to give a residue, which was purified by flash chromatography (1:1 hexane-EtOAc). 4-Alkoxypyridines were identified by ^1^H and ^13^C NMR spectroscopy.

#### 2.1.2. General Procedure B

Synthesis of TRIFAPYs (**2a–i**): 1,3,5-Tris(bromomethyl)benzene (1 equiv.) was added to a stirring solution of 4-alkoxypyridine (3 equiv.) in acetone or acetonitrile. The mixture was heated at reflux for 48 h and, after cooling to room temperature, the solvent was evaporated at reduced pressure. The crude was suspended in acetonitrile and stirred for 5 min. The resulting solid was separated by decantation, providing the corresponding pure TRIFAPY identified by ^1^H and ^13^C NMR spectroscopy.

### 2.2. Synthesis of Compounds

#### 2.2.1. 1,3,5-Tris[(4-butoxy-1-pyridinio)methyl]benzene Tribromide (2a)

Following the general procedure B, from 1,3,5-tris(bromomethyl)benzene (48 mg, 0.13 mmol) and 4-butoxypyridine (1a) [29](60 mg, 0.4 mmol) in acetonitrile (5 mL), TRIFAPY 2a (100 mg, 93%, MW 810.5 g/mol) was obtained. ^1^H NMR (400 MHz, CD_3_OD) δ 8.76 (sa, 6H), 7.64 (sa, 3H), 7.44 (d, *J* = 7.4 Hz, 6H), 5.61 (s, 6H), 4.29 (t, *J* = 6.5 Hz, 6H), 1.77 (sex, *J* = 7.1 Hz, 6H), 1.44 (sex, *J* = 7.7 Hz, 6H), and 0,91 (t, *J* = 7.4 Hz, 9H). ^13^C NMR (100.6 MHz, CD_3_OD) δ 171.2 (C), 146.1 (CH), 136.4 (C), 130.1 (CH), 113.9 (CH), 71.1 (CH_2_), 61.1 (CH_2_), 30.2 (CH_2_), 18.6 (CH_2_), and 12.6 (CH_3_). HRMS (ESI-TOF) *m*/*z*: Calculated for [C_36_H_48_N_3_O_3_]^+3^ 190.1226. Found 190.1233.

#### 2.2.2. 1,3,5-Tris[(4-hexyloxy-1-pyridinio)methyl]benzene Tribromide (2b)

Following the general procedure A, from 1-hexanol (1.78 mL, 14.2 mmol) in dry DMSO (11 mL), NaH (373 mg, 15.6 mmol, 90%), and crude 4-chloropyridine (1.6 g, 14.1 mmol), 4-alkoxypyridine **1b** [30] (1.11 g, 44%, MW 179.26 g/mol) was obtained. Following the general procedure B, from 1,3,5-tris(bromomethyl)benzene (150 mg, 0.42 mmol) and 4-hexyloxypyridine (225.5 mg, 1.26 mmol) in acetone (6 mL), TRIFAPY **2b** (328 mg, 88%, MW 891.22 g/mol) was obtained. ^1^H NMR (400 MHz, CD_3_OD) *δ* 8.88 (d, *J* = 7.2 Hz, 6H), 7.75 (s, 3H), 7.56 (d, *J* = 7.6 Hz, 6H), 5.75 (s, 6H), 4.40 (t, *J* = 6.4 Hz, 6H), 1.91 (sex, *J* = 6.4 Hz, 6H), 1.56–1.49 (m, 6H), 1.42–1.38 (m, 12H), and 0.95 (t, *J* = 7.2 Hz, 9H). ^13^C NMR (100.6 MHz, CD_3_OD) *δ* 171.2 (C), 146.1 (CH), 136.4 (C), 130.1 (CH), 113.9 (CH), 71.4 (CH_2_), 61.1 (CH_2_), 31.1 (CH_2_), 28.2 (CH_2_), 25.1 (CH_2_), 22.2 (CH_2_), and 12.9 (CH_3_). HRMS (ESI-TOF) *m*/*z*: Calculated for [C_42_H_60_N_3_O_3_]^+3^ 218.1539. Found 218.1550.

#### 2.2.3. 1,3,5-Tris[(4-octyloxy-1-pyridinio)methyl]benzene Tribromide (2c)

Following the general procedure A, from 1-octanol (1.68 mL, 10.62 mmol) in dry DMSO (9.6 mL), NaH (311 mg, 11.69 mmol, 90%), and crude 4-chloropyridine (1.20 g, 10.62 mmol), 4-alkoxypyridine **1c** [31,32] (0.42 g, 20%, MW 207.32 g/mol) was obtained. Following the general procedure B, from 1,3,5-tris(bromomethyl)benzene (57.47 mg, 0.161 mmol) and 4-octyloxypyridine (100 mg, 0.48 mmol) in acetone (14 mL), TRIFAPY **2c** (141 mg, 90%, MW 978.82 g/mol) was obtained. ^1^H NMR (400 MHz, CDCl_3_) *δ* 9.92 (d, *J* = 7.0 Hz, 6H), 8.99 (s, 3H), 7.34 (d, *J* = 7.0 Hz, 6H), 5.73 (s, 6H), 4.20 (t, *J* = 6.5 Hz, 6H), 1.85–1.81 (m, 6H), 1.44–1.38 (m, 6H), 1.34–1.27 (m, 24H), and 0.89 (t, *J* = 7 Hz, 9H). ^13^C NMR (100.6 MHz, CDCl_3_) *δ* 170.6 (C), 147.2 (CH), 135.5 (C), 132.9 (CH), 114.2(CH), 71.4 (CH_2_), 61.2 (CH_2_), 31.7 (CH_2_), 29.1 (CH_2_), 29.1 (CH_2_), 28.3 (CH_2_), 25.6 (CH_2_), 22.6 (CH_2_), and 14.1 (CH_3_). HRMS (ESI-TOF) *m*/*z*: Calculated for [C_48_H_72_N_3_O_3_]^+3^ 246.1852. Found 246.1852.

#### 2.2.4. 1,3,5-Tris[(4-decyloxy-1-pyridinio)methyl]benzene Tribromide (2d)

Following the general procedure A, from 1-decanol (2.53 mL, 13.27 mmol) in dry DMSO (12 mL), NaH (389 mg, 14.6 mmol, 90%), and crude 4-chloropyridine (1.5 g, 13.27 mmol), alkoxypyridine **1d** [12,31] (1.43 g, 46%, MW 235.37 g/mol) was obtained. Following the general procedure B, from 1,3,5-tris(bromomethyl)benzene (50.7 mg, 0.14 mmol) and 4-decyloxypyridine (100 mg, 0.43 mmol) in acetone (12.5 mL), TRIFAPY **2d** (139 mg, 92%, MW 1062.98 g/mol) was obtained. ^1^H NMR (400 MHz, CDCl_3_) *δ* 9.90 (d, *J* = 7.4 Hz, 6H), 8.96 (s, 3H), 7.36 (d, *J* = 7.4 Hz, 6H), 5.77 (s, 6H), 4.21 (t, *J* = 6.5 Hz, 6H), 1.87–1.80 (m, 6H), 1.44–1.40 (m, 6H), 1.30–1.27 (m, 36H), and 0.87 (t, *J* = 7 Hz, 9H). ^13^C NMR (100.6 MHz, CDCl_3_) *δ* 170.4 (C), 147.0 (CH), 135.4 (C), 132.6 (CH), 114.0 (CH), 71.3 (CH_2_), 61.0 (CH_2_), 31.6 (CH_2_), 29.3 (CH_2_), 29.2 (CH_2_), 29.1 (CH_2_), 28.9 (CH_2_), 28.2 (CH_2_), 25.4 (CH_2_), 22.5 (CH_2_), and 13.9 (CH_3_). HRMS (ESI-TOF) *m*/*z*: Calculated for [C_54_H_84_N_3_O_3_]^+3^ 274.2165. Found 274.2172.

#### 2.2.5. 1,3,5-Tris[(4-dodecyloxy-1-pyridinio)methyl]benzene Tribromide (2e)

Following the general procedure A, from 1-dodecanol (2.47 g, 13.27 mmol) in dry DMSO (8 mL), NaH (389 mg, 14.5 mmol, 90%), and crude 4-chloropyridine (1.5 g, 13.27 mmol), compound **1e** [31,32] (2.09 g, 60%, MW 263.43 g/mol) was obtained. Following the general procedure B, from 1,3,5-tris(bromomethyl)benzene (46.35 mg, 0.12 mmol) and 4-dodecyloxypyridine (100 mg, 0.38 mmol) in acetone (7 mL), TRIFAPY **2e** (63.4 mg, 44%, MW 1147.14 g/mol) was obtained. ^1^H NMR (400 MHz, CD_3_OD) *δ* 8.83 (d, *J* = 7.5 Hz, 6H), 7.68 (s, 3H), 7.56 (d, *J* = 7.6 Hz, 6H), 5.72 (s, 6H), 4.40 (t, *J* = 6.5 Hz, 6H), 1.95–1.87 (m, 6H), 1.56–1.49 (m, 6H), 1.43–1.28 (m, 48H), and 0.92 (t, *J* = 7.1 Hz, 9H). ^13^C NMR (100.6 MHz, CD_3_OD) *δ* 170.4 (C), 147.0 (CH), 135.5 (C), 132.6 (CH), 114.1 (CH), 71.3 (CH_2_), 61.1 (CH_2_), 31.8 (CH_2_), 29.5 (CH_2_), 29.4 (CH_2_), 29.3 (CH_2_), 29.2 (CH_2_), 29.0 (CH_2_), 28.3 (CH_2_), 25.5 (CH_2_), 22.6 (CH_2_), and 14.0 (CH_3_). HRMS (ESI-TOF) *m*/*z*: Calculated for [C_60_H_96_N_3_O_3_]^+3^ 302.2478. Found 302.2465.

#### 2.2.6. 1,3,5-Tris[(4-tetradecyloxy-1-pyridinio)methyl]benzene Tribromide (2f)

Following the general procedure A, from 1-tetradecanol (2.17 g, 10.61 mmol) in dry DMSO (8 mL), NaH (311 mg, 11.67 mmol, 90%), and crude 4-chloropyridine (1.2 g, 10.61 mmol), alkoxypyridine **1f** [31] (2.16 g, 70%, MW 291.48 g/mol) was obtained. Following the general procedure B, from 1,3,5-tris(bromomethyl)benzene (12.6 mg, 0.034 mmol) and 4-tetradecyloxypyridine (**1f**, 30 mg, 0.10 mmol) in acetone (3 mL), TRIFAPY **2f** (17.7 mg, 41%, MW 1231.31 g/mol) was obtained. ^1^H NMR (400 MHz, CD_3_OD) *δ* 8.82 (d, *J* = 7.4 Hz, 6H), 7.66 (s, 3H), 7.56 (d, *J* = 7.5 Hz, 6H), 5.71 (s, 6H), 4.40 (t, *J* = 6.5 Hz, 6H), 1.91 (sex, *J* = 6.8 Hz, 6H), 1.56–1.49 (m, 6H), 1.42–1.29 (m, 60H), and 0.92 (t, *J* = 7.0 Hz, 9H). ^13^C NMR (100.6 MHz, CDCl_3_) *δ* 170.5 (C), 147.2 (CH), 135.5 (C), 132.9 (CH), 114.2 (CH), 71.5 (CH_2_), 61.2 (CH_2_), 31.9 (CH_2_), 29.7 (CH_2_), 29.6 (CH_2_), 29.6 (CH_2_), 29.5 (CH_2_), 29.5 (CH_2_), 29.4 (CH_2_), 29.3 (CH_2_), 29.1 (CH_2_), 28.4 (CH_2_), 25.6 (CH_2_), 22.7 (CH_2_), and 14.1 (CH_3_). HRMS (ESI-TOF) *m*/*z*: Calculated for [C_66_H_108_N_3_O_3_]^+3^ 330.2791. Found 330.2802.

#### 2.2.7. 1,3,5-Tris[(4-hexadecyloxy-1-pyridinio)methyl]benzene Tribromide (2g)

Following the general procedure A, from 1-hexadecanol (3.68 g, 15.18 mmol) in dry DMSO (26 mL), NaH (444 mg, 16.7 mmol, 90%), and crude 4-chloropyridine (1.72 g, 15.18 mmol), room temperature for 24 h, alkoxypyridine **1g** [32] (2.32 g, 48%, MW 319.53 g/mol) was obtained. Following the general procedure B, from 1,3,5-tris(bromomethyl)benzene (74.6 mg, 0.21 mmol) and 4-hexadecyloxypyridine (**1g**) (200 mg, 0.62 mmol) in acetone (18 mL), TRIFAPY **2g** (36 mg, 13%, MW 1315.47 g/mol) was obtained after flash chromatography (6:4 CH_2_Cl_2_-MeOH). ^1^H NMR (400 MHz, CDCl_3_) *δ* 9.96 (d, *J* = 7 Hz, 6H), 9.04 (s, 3H), 7.32 (d, *J* = 7 Hz, 6H), 5.71 (s, 6H), 4.19 (t, *J* = 6.5 Hz, 6H), 1.79–1.86 (m, 6H), 1.48–1.35 (m, 6H), 1.32–1.26 (m, 72H), and 0.87(t, *J* = 6.5 Hz, 9H). ^13^C NMR (100.6 MHz, CDCl_3_) *δ* 170.6 (C), 147.2 (CH), 135.4 (C), 133.0 (CH), 114.1 (CH), 71.4 (CH_2_), 61.2 (CH_2_), 31.9 (CH_2_), 29.7 (CH_2_), 29.6 (CH_2_), 29.6 (CH_2_), 29.5 (CH_2_), 29.4 (CH_2_), 29.3 (CH_2_), 29.1 (CH_2_), 28.3 (CH_2_), 25.6 (CH_2_), 22.7 (CH_2_), and 14.1 (CH_3_). HRMS (ESI-TOF) *m*/*z*: Calculated for [C_72_H_120_N_3_O_3_]^+3^ 358.3104. Found 358.3108.

#### 2.2.8. 1,3,5-Tris[(4-octadecyloxy-1-pyridinio)methyl]benzene Tribromide (2h)

Following the general procedure A, from 1-octadecanol (2.5 g, 9.24 mmol) in dry DMSO (18 mL), NaH (277 mg, 10.44 mmol, 90%), and crude 4-chloropyridine (1.18 g, 10.44 mmol), alkoxypyridine **1h** [12] (1.94 g, 53%, MW 347.59 g/mol) was obtained. Following the general procedure B, from 1,3,5-tris(bromomethyl)benzene (34.3 mg, 0.096 mmol) and 4-octadecyloxypyridine (**1h**, 100 mg, 0.29 mmol) in acetone (8.5 mL), TRIFAPY **2h** (69 mg, 51%, MW 1399.63 g/mol) was obtained after flash chromatography (6:4 CH_2_Cl_2_-MeOH). ^1^H NMR (400 MHz, CD_3_OD) *δ* 8.75 (d, *J* = 7.5 Hz, 6H), 7.61 (s, 3H), 7.43 (d, *J* = 7.6 Hz, 6H), 5.62 (s, 6H), 4.28 (t, *J* = 6.5 Hz, 6H), 1.78 (quint, *J* = 6.8 Hz, 6H), 1.44–1.36 (m, 6H), 1.31–1.25 (m, 84H), and 0.80 (t, *J* = 7.0 Hz, 9H). ^13^C NMR (100.6 MHz, CD_3_OD) *δ* 172.6 (C), 147.4 (CH), 137.8 (C), 131.4 (CH), 115.3 (CH), 72.8 (CH_2_), 62.5 (CH_2_), 33.1 (CH_2_), 30.8 (CH_2_), 30.8 (CH_2_), 30.7 (CH_2_), 30.7 (CH_2_), 30.5 (CH_2_), 30.4 (CH_2_), 29.6 (CH_2_), 26.8 (CH_2_), 23.7 (CH_2_), and 14.4 (CH_3_).

#### 2.2.9. 1-Eicosanol (3)

LiAlH_4_ (24 mL, 1M in THF, 23.99 mmol) was added to a stirring solution of arachidic acid (2.5 g, 7.99 mmol) in anhydrous THF (22.5 mL) at 0 °C and the mixture was stirred at room temperature for 3 h. Water was slowly added, and the resulting mixture was diluted with EtOAc. The phases were separated, and the aqueous solution was extracted with EtOAc. The combined organic extracts were dried, filtered, and concentrated, providing alcohol **3** as an off-white solid (801 mg, 34%, MW 298.56 g/mol). IR: 3332 (OH) cm^−1^. ^1^H NMR (400 MHz, CDCl_3_) *δ* 3.64 (t, *J* = 6.6 Hz, 2H), 1.60–1.53 (m, 2H), 1.35–1.23 (m, 34H), and 0.88 (t, *J* = 6.7 Hz, 3H). ^13^C NMR (100.6 MHz, CDCl_3_) *δ* 63.1 (CH_2_), 32.8 (CH_2_), 31.9 (CH_2_), 29.7 (CH_2_), 29.7 (CH_2_), 29.6 (CH_2_), 29.6 (CH_2_), 29.4 (CH_2_), 29.4 (CH_2_), 25.7 (CH_2_), 22.7 (CH_2_), and 14.1 (CH_3_).

#### 2.2.10. 1,3,5-Tris[(4-eicosanyloxy-1-pyridinio)methyl]benzene Tribromide (2i)

Following the general procedure A, from 1-eicosanol (1.05 g, 3.52 mmol) in dry DMSO (7.2 mL), NaH (103 mg, 3.87 mmol, 90%), and crude 4-chloropyridine (0.39 g, 3.52 mmol), alkoxypyridine **1i** (40 mg, 3%, MW 375.64 g/mol) was obtained. ^1^H NMR (400 MHz, CDCl_3_) *δ* 8.34 (d, *J* = 6.0 Hz, 2H), 6.73 (d, *J* = 6.0 Hz, 2H), 3.96 (t, *J* = 6.6 Hz, 2H), 1.76–1.69 (m, 2H), 1.41–1.34 (m, 2H), 1.29–1.15 (m, 32H), and 0.80 (t, *J* = 7 Hz, 3H). ^13^C NMR (100.6 MHz, CDCl_3_) *δ* 165.3 (C), 150.7 (CH), 110.3 (CH), 68.0 (CH_2_), 31.9 (CH_2_), 29.7 (CH_2_), 29.6 (CH_2_), 29.6 (CH_2_), 29.5 (CH_2_), 29.5 (CH_2_), 29.3 (CH_2_), 29.3 (CH_2_), 28.8 (CH_2_), 25.9 (CH_2_), 22.7 (CH_2_), and 14.1 (CH_3_). Following the general procedure B, from 1,3,5-tris(bromomethyl)benzene (7.9 mg, 0.022 mmol) and 4-eicosanyloxypyridine (25 mg, 0.067 mmol) in acetone (2 mL), TRIFAPY **2i** (6.3 mg, 19%, MW 1483.79 g/mol) was obtained after flash chromatography (6:4 CH_2_Cl_2_-MeOH). ^1^H NMR (400 MHz, CDCl_3_) *δ* 9.93 (sa, 6H), 9.02 (sa, 3H), 7.33 (sa, 6H), 5.70 (s, 6H), 4.18 (t, *J* = 6.5 Hz 6H), 1.80–1.78 (m, 6H), 1.45–1.37 (m, 6H), 1.35–1.21 (m, 96H), and 0.87 (t, *J* = 7.0 Hz, 9H). ^13^C NMR (100.6 MHz, CDCl_3_) *δ* 170.6 (C), 147.3 (CH), 135.4 (C), 133.0 (CH), 114.2 (CH), 71.4 (CH_2_), 61.2 (CH_2_), 31.9 (CH_2_), 29.7 (CH_2_), 29.7 (CH_2_), 29.6 (CH_2_), 29.6 (CH_2_), 29.5 (CH_2_), 29.5 (CH_2_), 29.4 (CH_2_), 29.4 (CH_2_), 29.2 (CH_2_), 29.1 (CH_2_), 28.4 (CH_2_), 25.6 (CH_2_), 22.7 (CH_2_), and 14.1 (CH_3_). HRMS (ESI-TOF) *m*/*z*: Calculated for [C_84_H_144_N_3_O_3_]^+3^ 414.373. Found 414.3719.

### 2.3. Design and Usage of PPRH and Antisense Oligonucleotide

The search for the polypurine sequences that conform to the hairpin structure of the PPRH was accomplished by the Triplex-forming Oligonucleotide Target Sequence Search software (http://utw10685.utweb.utexas.edu/tfo/ (accessed on 23 March 2024) MD Anderson cancer center, The University of Texas) [33]. We previously validated in our laboratory the usage of the HpsPr-C PPRH against the promoter sequence of survivin to specifically produce gene silencing of this antiapoptotic protein by decreasing its levels of mRNA and protein [26,28]. The antisense oligonucleotide was directed to the first 21 nucleotides after the initiation of translation of survivin [34]. An unspecific scrambled hairpin (HpScr9), which was unable to form a triplex structure with the target DNA, was used as the negative control. For the internalization experiments, this PPRH was labeled with fluorescein (FAM) on its 5′ end All oligonucleotides employed in this study were synthesized as non-modified oligodeoxynucleotides obtained from Merck (Haverhill, UK), resuspended in sterile Tris-EDTA buffer (1 mM EDTA and 10 mM Tris, pH 8.0) (Merck, Madrid, Spain), and stored at −20 °C until use. The PPRH sequences and that of the antisense oligonucleotide are shown in Table 1.

### 2.4. Agarose Gel Retardation Assays

Binding reactions containing 100 ng of FAM-HpScr9 (scramble) and increasing amounts of the transfection agents were incubated in a final volume of 10 μL. After 20 min at room temperature, 2 μL of loading buffer 6× was added to the samples, which were subsequently subjected to electrophoresis in 0.8% agarose gels in TAE buffer 1× (40 mM Tris-acetate, 1 mM EDTA, pH 8.0). Gels were visualized with UV light in a Gel Doc^TM^ EZ (Bio-Rad Laboratories, Inc., Barcelona, Spain).

### 2.5. Characterization of Complexes by DLS and Cryo-TEM

The size of the nanoparticle formed by the PPRH and TRIFAPY C12 was assessed using dynamic light scattering (DLS) by a Zetasizer Nano (Malvern, UK) at a constant angle of 90°. The preparation of the liposome together with the PPRH for the measurements was performed under the same ratios employed in the transfection process.

For cryo-TEM observations, grids were transferred to a Tecnai F20 (FEI, Eindhoven, the Netherlands) using a cryoholder (Gatan, Warrendale, PA, USA). Images were taken at 200 kV, at a temperature ranging from −175 to −170 °C, and using low-dose imaging conditions with a 4096 × 4096 pixel CCD Eagle camera (FEI, Eindhoven, the Netherlands).

### 2.6. Cell Culture

The PC-3 prostate cancer and SKBR-3 breast cancer cell lines were obtained from the Cell Bank resources of the University of Barcelona. Cells were grown in Ham’s F12 medium with 10% fetal bovine serum and were incubated at 37 °C in a humified atmosphere at 5% CO_2_. The hCMEC/D3 cell line was grown in Endothelial Cell Basal Medium-2 (EBM^®^-2, from Lonza, Walkersville, MD, USA). Subculture was performed with Trypsin 0.05% (Merck, Madrid, Spain).

### 2.7. Transfection of PPRHs

Next, 24 h before transfection, cells were plated in 6-well dishes in F12 medium. The transfection mix contained different amounts of the corresponding TRIFAPY and 100 nM of the PPRH in a serum-free medium up to 100 µL. The mix was incubated for 20 min to facilitate the formation of the DNA-TRIFAPY complex and was added to the cells to a final volume of 1 mL of Ham’s F12 medium.

### 2.8. MTT Assays

PC-3 (10,000) and SKBR-3 (30,000) cells were plated in 6-well dishes in F12 medium. hCMEC/D3 (30,000) cells were grown in EBM^®^-2. TRIFAPYs were co-incubated with the oligonucleotide and added to the cells after 20 min. Five days later, 0.63 mM of 3-(4,5-dimetilthyazol-2-yl)-2,5-diphenilte-trazolium bromide and 100 µM sodium succinate (both from Merck, Madrid, Spain) were added to each well. After 3 h at 37 °C of incubation, culture media were aspirated and replaced by a lysis solution (0.57% acetic acid and 10% sodium dodecyl sulfate in dimethyl sulfoxide) (Merck, Madrid, Spain). Absorbance at 570 nm was measured in a Varioskan Lux, (Thermo Scientific, Barcelona, Spain). Cell viability was calculated as the percentage of cell survival, relative to the control samples.

### 2.9. Cellular Uptake

The internalization of the complex into PC-3 and hCMEC/D3 cells was monitored by fluorescence microscopy and flow cytometry. In total, 90,000 cells were plated in 6-well dishes in F12 medium or EBM^®^-2, respectively. The following day, incubations with 100 nM PPRH labeled with fluorescence (FAM-HpScr-9) and 1.5 µg/mL of the corresponding TRIFAPYs were performed for 24 h. Firstly, images were captured by fluorescence microscopy (ZOE Fluorescent Cell Imager, Bio-Rad Laboratories, Inc., Spain). Then, cells were trypsinized and collected in cold PBS. The suspension was centrifuged at 800× *g* at 4 °C for 5 min, and the cell pellet was resuspended in 400 μL of PBS. Propidium iodide was added to achieve a final concentration of 5 μg/mL (Merck, Madrid, Spain), and samples were analyzed using a Gallios flow cytometer (Beckman Coulter, Inc., Barcelona, Spain) from the Scientific and Technological Centers of the University of Barcelona (CCiTUB).

To investigate the internalization mechanism of the complexes formed by PPRH and TRIFAPY, prostate cancer cells PC-3 (120,000) were plated in 6-well dishes in the F12 medium. After 24 h, cells were incubated with different inhibitors: 75 µM of Dynasore [35] to hinder clathrin-dependent endocytosis; 185 µM of Genistein [36]to block caveolin-mediated endocytosis; or 33 µM of 5-(N-ethyl-N-isopropyl) amiloride (EIPA) [37] to inhibit macropinocytosis, all of them purchased from Merck (Madrid, Spain). Inhibitors were incubated for 60 min at 37 °C. Then, cells were transfected with FAM-HpScr-9 and TRIFAPY C12 for 3.5 h. Subsequently, cells underwent flow cytometry analyses.

### 2.10. Apoptosis Assays

Cells (100,000) were incubated with the transfection agent and PPRH (HpsPr-C) for 48 h. Then, cells were trypsinized, collected in PBS, and centrifuged at 1200× *g* for 5 min. The resulting pellet was resuspended in 100 μL of Binding Buffer 1× and incubated with 5 μL of the APC Annexin V Detection Kit (Invitrogen, Thermo Scientific, Barcelona, Spain) for 15 min at room temperature. After a 5 min centrifugation at 1200× *g*, the pellet was washed with 1 mL of Binding Buffer 1×. The samples were centrifuged for 5 min, resuspended in a final volume of 500 μL of Binding Buffer 1×, and propidium iodide was added to achieve a concentration of 5 μg/mL (Merck, Madrid, Spain). Apoptosis analyses were performed using a Gallios flow cytometer (Beckman Coulter, Inc., Barcelona, Spain) from the CCiTUB.

### 2.11. Survivin mRNA Levels

RNA from control and treated cells after 24 h of transfection were extracted using Trizol Reagent (Life Technologies, Madrid, Spain). Synthesis of the complementary DNA was performed in a final volume of 20 µL reaction mixture containing 1 µg of RNA, 0.5 mM of each deoxyribonucleotide triphosphate (dNTP, Epicentre, Madison, WI, USA), 250 ng of random hexamers (Roche, Barcelona, Spain), 10 mM dithiothreitol, 200 units of a Moloney murine leukemia virus reverse transcriptase (RT), 20 units of RNase inhibitor, and 4 μL of buffer (5×) (all three from Lucigen, Middleton, WI, USA). The reaction was incubated at 42 °C for 1 h. Once the cDNA was obtained, we proceeded to the preparation of the RT-qPCR mixture carried out in 20 µL containing 1× TaqMan Universal PCR Master mix (Applied Biosystems, Madrid, Spain), 0.5× TaqMan probe, and 3 μL of cDNA. To determine survivin mRNA levels, the BIRC5 mRNA TaqMan probe (Hs04194392_s1; ThermoFisher Scientific, Madrid, Spain) was used, with Cyclophilin PP1A mRNA (Hs04194521_s1, Thermo Fisher Scientific, Madrid, Spain) as the endogenous control. The RT-qPCR cycling conditions were 10 min of denaturation at 95 °C, then 40 cycles of 15 s at 95 °C, and 1 min at 60 °C using a QuantStudio 3 Real-time PCR System (Applied Biosystems, Barcelona, Spain). Quantification was performed using the ΔΔCt method, where Ct is the threshold cycle that corresponds to the cycle when the amount of amplified mRNA reaches the fluorescence threshold.

## 3. Results

### 3.1. Synthesis of TRIFAPYs

The synthesis of the tricationic compounds was performed as described in Materials and Methods. To explore the influence of the length of the alkyl chain linked to the pyridinium cation on the biological activity associated with this core, alkoxypyridines **1a–i** (Figure 1) were prepared by an aromatic nucleophilic substitution reaction from 4-chloropyridine and the corresponding alkoxide.

Then, a nucleophilic substitution reaction at the three benzylic positions of 1,3,5-tris(bromomethyl)benzene afforded the corresponding tricationic compounds **2a–i** (Figure 2). The structure of TRIFAPYs was unequivocally confirmed by ^1^H NMR, ^13^C NMR, and HRMS (Appendix A).

### 3.2. Characterization of Complexes Formed by TRIFAPYs and PPRH

The capability of TRIFAPYs to interact with FAM-HpScr9 PPRH was evaluated by gel retardation assays (Figure 3A). The binding reactions were set up with a fixed amount of the fluorescent PPRH and increasing amounts of TRIFAPYs. Transfection agent concentrations varied depending on the binding ability of each of the compounds assessed. The band in the first lane in the agarose gels corresponds to the fluorescent intensity produced by the free PPRH. This band gradually decreased as the quantity of lipidic agent in the reaction increased, allowing for the determination of the K_D_ of each TRIFAPY, as shown in Figure 3A. The hydrodynamic diameter of the complex formed by the PPRH and TRIFAPY C12 was 125 nm (±1.33) with a polydispersion of 0.105 (±2.08), both analyzed by DLS (Figure 3B). Complexes formed by the union of the PPRH and TRIFAPY C12 were visualized as well-defined concentrical vesicles by Cryo-transmission Electron Microscopy captures (Figure 3C).

### 3.3. Intrinsic Toxicity and Transfection Efficiency of TRIFAPYs

Cell viability was assessed in PC-3 cells after transfection of HpsPr-C with all the synthesized TRIFAPYs and DOTAP (Figure 4A). While the TRIFAPYs did not show significant toxicity by themselves at 1.5 µg/mL, the compounds containing alkyl chains from C12 through C18 showed a clear effect in decreasing cell viability when combined with the PPRH. However, compounds C4 through C10 did not cause any significant decrease in viability. Thus, TRIFAPYs from C12 to C18 were selected for further experiments. These transfectant agents were also subjected to an analysis at higher concentrations to establish a working range (Figure 4B). Furthermore, the transfection efficiency of these molecules was also assessed in the breast cancer cell line SKBR-3 using a fixed concentration for all of them (Figure 4C). None of the tested TRIFAPYs induced a significant reduction in cell viability individually in this cell line either. Nevertheless, they exhibited a significant decline in viability upon combination with HpsPr-C. Finally, the hCMEC/D3 cell line was employed as a model for the blood–brain barrier to investigating the efficiency of these lipidic agents in transfecting therapeutic oligonucleotides (Figure 4D). Since only TRIFAPY C12 could be classified within the category of medium-chain fatty acids (MCFA), it was investigated as a candidate to potentially transfect blood–brain barrier (BBB) cells. Using different concentrations of TRIFAPY C12, a notable decrease in cell viability was observed upon the transfection of HpsPr-C using this agent, whereas TRIFAPY C12 itself showed no toxicity in a range of concentrations. The effect of TRIFAPY C18 was also studied for comparison purposes since the size of its alkyl chain corresponds to a long-chain fatty acid (LCFA) and, thus, is not expected to permeate BBB cells. No differences in cell viability were observed between cells treated with C18 alone and in combination with HpsPr-C as anticipated. Finally, cytotoxicity assays performed using TRIFAPY C12 for the transfection of the survivin antisense oligonucleotide resulted in a reduction in cell viability at different concentrations in the PC-3 cell line (Figure 4E). As was previously shown [34], the effective concentration of the antisense oligonucleotide had to be higher (800 nM) than those employed with PPRH.

### 3.4. Cellular Uptake of PPRH into Cancer Cell Lines

The internalization of the complex formed by TRIFAPYs and the PPRH into prostate cancer cells PC-3 and in hCMEC/cells was studied through fluorescence microscopy and flow cytometry analyses. Cellular uptake was assessed 24 h upon incubations with a green fluorescent labeled PPRH (FAM-HpScr9) in combination with the previously selected transfectant agents. All four candidates and DOTAP were tested in PC-3, while in hCMEC/D3, incubations were performed using TRIFAPY C12 and C18. Images captured by fluorescence microscopy (ZOE Fluorescent Cell Imager Bio Laboratories) showed green fluorescence in the cells in all cases for PC-3 (Figure 5). However, in hCMEC/D3, only TRIFAPY C12 was capable of internalizing the fluorescent complex (Figure 6). Flow cytometry analysis results were presented in four quadrants. When compared to untreated control PC-3 cells, the incubation of the PPRH with TRIFAPYs C12, C14, C16, and C18 caused a shift to the right of the cell population, indicating an increase in positive gated cells (Table 2). However, in hCMEC/D3, TRIFAPY C12 caused a higher percentage of positive gated cells than TRIFAPY C18 (Table 3) given the length of alkyl chains. Additionally, the parameter X-MEAN, corresponding to the intensity of fluorescence taken by the cells, was determined in both assays. It is noteworthy that the incubations did not exhibit any effect on cell viability, as the scramble PPRH was non-specific.

The mechanisms involved in complex internalization were also studied by flow cytometry, using the endocytosis inhibitors Dynasore, Genistein, and EIPA during the incubation of the PC-3 cell line with FAM-HpScr9 and TRIFAPY C12 (Figure 7). Upon comparison of the fluorescence observed in cells treated with the fluorescent PPRH in the absence of inhibitors, Dynasore induced a substantial reduction (83.3%) in cells showing positive internalization (Table 4). This effect was also reproduced in the X-MEAN parameter, indicating that the main entry pathway for the complex was clathrin-mediated endocytosis. In addition, the complex could be secondarily internalized by caveolin-mediated endocytosis and macropinocytosis.

### 3.5. Effect on Survivin mRNA Levels and Apoptosis Caused by Therapeutic Oligonucleotides Transfected with TRIFAPYs

Once the internalization of the complex was confirmed using a non-specific scrambled PPRH, we proceeded to study the mRNA levels of survivin in the PC-3 cell line after the incubation of HpsPr-C or aODN-Survivin. Both oligonucleotides were designed to target the gene encoding for survivin, an antiapoptotic protein implicated in the proliferation of tumors. mRNA levels were analyzed by RT-qPCR after the transfection of HpsPr-C in combination with the commercialized liposome DOTAP or with the family of TRIFAPYs from C12 to C18 (Figure 8A), which produced a significant reduction in survivin mRNA levels. Furthermore, a significant decrease was also observed upon the transfection of antisense oligonucleotide at 800 nM when using TRIFAPY C12 (Figure 8B).

To assess the effect on apoptosis, PC-3 cells were treated with HpsPr-C in combination with the same TRIFAPYs or with DOTAP. Apoptosis levels were analyzed by the Annexin V method using flow cytometry. In comparison to untreated cells, the population of transfected cells showed an increase in the levels of apoptosis (Figure 9) (Table 5).

## 4. Discussion

The development of gene therapies holds great significance in studying cellular and molecular processes with potential therapeutic applications for treating genetic diseases. However, the limitation of these practices lies in the lack of efficient transfection systems. Delivery agents enable the carry of therapeutics nucleic acids, such as PPRHs, into the cell nucleus to regulate gene expression. These systems must provide protection against degradation mediated by nucleases or the physicochemical conditions of the environment, ensuring greater stability, specificity, and safety.

This multidisciplinary project combines elements of Organic Chemistry, Biochemistry, and Molecular Biology by synthesizing, characterizing, and validating new transfectant agents using PPRHs as gene-silencing tools. Previously, we developed two liposomes, namely DOPY 1,3-bis [(4-oleyloxy-1-pyridinio)methyl]benzene dibromide and TROPY 1,3,5-tris[(4-oleyloxy-1-pyridinio)methyl]benzene tribromide. In this work, we proceeded to synthesize a complete series of cationic transfectant agents [38,39], which are pyridinium-based with three molecules of fatty acids (TRIFAPYs) varying in the length of carbon units ranging from C4 to C20, testing their ability to serve as transfecting agents and using PPRHs as a model of therapeutic oligonucleotides. Since PPRHs are formed by non-modified bases and exhibit a short length, they present very low immunogenicity and lack of hepatotoxicity or nephrotoxicity. Furthermore, polypurine hairpins demonstrated notable efficacy compared to ASOs, as they can be employed at concentrations ten times lower. The intramolecular Reverse Hoogsteen bonds account for the high stability of PPRHs at physiological pH. Lastly, it is important to note that their synthesis is cost-effective.

The transfection agents were synthesized in a two-step process starting from the corresponding alkyl alcohol, 4-chloropyridine, and 1,3,5-tris(bromomethyl)benzene. The alkoxypyridines **1a–i** were obtained in the first step through an aromatic nucleophilic substitution reaction between the corresponding alkoxide and the pyridine. In the second step, the corresponding tricationic compounds **2a–i** were obtained by a bimolecular nucleophilic substitution reaction. Yields ranged from excellent (80–95%) to moderate (40–50%) and very low (around 10%) depending on the length of the alkyl chain. Except for 1-eicosanol, all alkyl alcohols used in this study are commercially available. Thus, prior to the preparation of alkoxy-pyridine **1i**, 1-eicosanol was synthesized by the reduction of arachidic acid with aluminum hydride. The structure of all compounds described in this study (**1a–i**, **2a–i**, and **3**) has been characterized and confirmed through ^1^H NMR, ^13^C NMR spectroscopy, and high-resolution mass spectrometry (HRMS).

The obtained compounds were dissolved in DMSO. It was noted that as the number of carbons in the lipid chain of TRIFAPYs increased, the solubility of the compounds decreased. Specifically, compound **2i** (also referred to as TRIFAPY C20) became insoluble in DMSO. Consequently, its efficacy validation was unattainable.

TRIFAPYs are tricationic compounds with the ability to establish electrostatic interactions with the negative charges present in the DNA phosphodiester bonds, as observed in the gel retardation assays. However, each TRIFAPY compound exhibited a different ability to bind PPRHs as evidenced by the K_D_ parameter. Compounds with shorter fatty acid chains (C4 and C6), as well as TRIFAPYs with longer chains (C16 and C18), had a lower affinity to bind DNA, showing K_D_ values above 1 µg/µL. Conversely, TRIFAPYs with a medium chain length ranging between 8 and 14 carbons exhibited higher affinities for PPRH, resulting in K_D_ values below 0.1 µg/µL. We conclude that the length of fatty acids may slightly alter the overall structure of the compounds, leading to different access to the cationic charges in the pyridinium ring. Consequently, the quantity of transfectants in the gel retardation assays was adjusted individually for each compound. Complexes were visualized by Cryo-TEM as well-defined round vesicles.

An initial cytotoxicity assay was performed using the synthesized compound collection of TRIFAPYs in PC-3 cells. The selection of the prostate cancer cell line PC-3 for this screening is justified by prior gene silencing studies performed in our research group using HpsPr-C transfected with commercially available liposome DOTAP, which resulted in a reduction in cell viability of 85%, whereas it was non-toxic on its own [12,28]. TRIFAPYs from C12 to C18 (**2e–h**) did not decrease cell viability by themselves but caused a significant reduction when incubated with HpsPr-C. However, TRIFAPYs C4 to C10 (**2a–d**) showed no effects on cell viability either alone or in combination with the PPRH. Thus, the series of TRIFAPYs C12 through C18 were selected for upcoming experiments. The original work with PC-3 cells was expanded to the SKBR-3 cell line, opening the usage of these new agents for both sexes, as is the case with prostate and breast cancers. In SKBR-3 cells, it was determined that none of the selected TRIFAPYs were toxic by themselves, while significant differences were observed when transfecting the cells with HpsPr-C. Furthermore, additional cytotoxicity assays were conducted in PC-3, resulting in a reduction in cellular viability upon the transfection of an antisense oligonucleotide designed against survivin when using TRIFAPY C12.

After performing cytotoxicity assays, it was observed that one of the selected TRIFAPYs, C12, could be categorized as a medium-chain fatty acid (MCFA), which consists of chains ranging between 6 and 12 carbon atoms. On the other hand, TRIFAPYs C14, C16, and C18 are classified within the long-chain fatty acids (LCFA), which encompass those with more than 12 carbon atoms. MCFAs, including lauric acid (C:12), are distinguished by their rapid absorption compared to LCFAs, as they do not require membrane transporters and can directly access the intracellular milieu [40]. Furthermore, they can penetrate the mitochondrial inner membrane without the need for the carnitine shuttle. Several in vivo studies in animals have reported that these MCFAs can swiftly traverse the blood–brain barrier (BBB) or undergo oxidation in the brain. Therefore, we aimed to explore the potential of TRIFAPY C12 in combination with a specific PPRH to cross the BBB. To test this hypothesis, the hCMEC/D3 cell line was used as a model for the human blood–brain barrier [41]. Various molecular assays have indicated that this cell line exhibits restricted permeability to certain molecules, specifically due to the presence of distinct efflux transporters (ABC) that actively extrude substrates, such as P-glycoprotein and multidrug resistance-associated proteins MRP-4 and MRP-5. Moreover, it has been reported that this cell line also expresses influx transporters (SLC) as insulin receptors, LDL receptors or Glut-1, which allow the entrance of glucose and amino acids. Regarding endothelial cell junctions, hCMEC/D3 cells express JAM-A, VE-cadherin, claudin-3, claudin-5, PECAM-1, occludins and zonula occludens protein 1 and 2. This similarity with the BBB led to the utilization of hCMEC/D3 as a representative model for the study of the capability of the nanoparticle complex formed by the transfectant agent and the oligonucleotide to cross the BBB. A dose–response analysis was performed in hCMEC/D3 using TRIFAPYs C12 and C18 alone and in combination with HpsPr-C. For C12, it was determined a non-significative decrease in cell viability by itself. However, a significant effect was observed upon its combination with HpsPr-C. On the other hand, no differences were observed in the incubations conducted with TRIFAPY C18 in combination with the same PPRH, as anticipated, since its alkyl chains belong to the LCFA category. In further approaches, it would be of interest to test C12 for the transfection of specific PPRHs directed to relevant targets involved in the treatment of neurological diseases.

The internalization of the fluorescent complex formed by FAM-HpScr9 and TRIFAPYs in PC-3 and hCMEC/D3 cells was detected both by fluorescence microscopy and flow cytometry. In PC-3, selected TRIFAPYs C12 to C18 caused positive gated cell values ranging from 50% to 70%, indicating the entrance of labeled PPRH into the cells mediated by the transfectant agents. These results were similar to those reported using DOTAP, resulting in a percentage of internalization up to 69% [42]. Nevertheless, it is worth mentioning that the working concentrations used for the TRIFAPYs were in the range of 1 µM, which is 10 times lower than those used for DOTAP, at 10 µM [13,42]. However, as was previously anticipated by the cytotoxicity experiments performed in hCMEC/D3, TRIFAPY C12 caused a higher effect in cells from the BBB model in comparison with incubations with TRIFAPY C18. The main pathway involved in the cellular uptake of the complex formed by PPRH and TRIFAPY is endocytosis mediated by clathrin.

As a model of therapeutic oligonucleotide in this study, we used the specific PPRH HpsPr-C validated with other liposomes such as DOTAP [28], DOPY [12] or TROPY [13] in the prostate cancer cell line PC-3. This PPRH targets the gene encoding for survivin, an antiapoptotic protein overexpressed in certain cancers [43,44] such as prostate [45], breast [46,47], neuroblastoma [47,48], and osteosarcoma [49]. Accordingly, we determined a reduction of 50% in mRNA levels of BIRC5 upon the transfection of HpsPr-C with TRIFAPYs C12 to C18. Moreover, the transfection with TRIFAPYs of this PPRH caused a noticeable increase in apoptosis in all incubations. Results reported using DOTAP at 10 µM for the transfection of HpsPr-C showed the same reduction in mRNA levels and an increment in apoptosis of 55% after the treatment. To validate the usage of these molecules for the transfection of other therapeutic oligonucleotides, the antisense oligonucleotide aODN-Survivin, previously used in cytotoxicity analysis, caused a 50% decrease in BIRC5 mRNA levels at the highest concentration. Therefore, both oligonucleotides produced a gene silencing effect when delivered by these novel tricationic agents.

## 5. Conclusion

This study describes the synthesis of TRIFAPYs, a collection of lipid-based compounds, and their validation as a potential transfection agent using the PPRH tool. The basic chemical structure of these agents is based on a pyridinium ring attached to alkyl residues of different lengths, enabling electrostatic interaction with PPRHs forming defined vesicles. They showed a high binding affinity, as shown by their low K_D_. TRIFAPYs C12 to C18 produced a high reduction in viability in PC-3 cells upon the transfection of HpsPr-C. The effect caused by the PPRH was also observed in the breast cancer cell line SKBR-3, proving the effectiveness of these new transfectants for both sexes. TRIFAPY C12 was also able to transfect efficiently HpsPr-C into the hCMEC/D3 cell line, a model for the BBB. Cellular uptake of the complex formed by HpScr-9 and TRIFAPYs C12 to C18 was confirmed by fluorescence microscopy and flow cytometry, and, by using specific transport inhibitors, it was determined that clathrin-mediated endocytosis was the principal pathway involved in the entry of the complexes. Finally, the incubation of the PPRH targeting survivin HpsPr-C with these TRIFAPYs reduced the mRNA levels of BIRC5 encoding for the antiapoptotic protein survivin, whereas apoptosis levels increased. In conclusion, we designed and synthesized TRIFAPYs C12, C14, C16, and C18 and validated them for the transfection of therapeutic oligonucleotides into mammalian cells including the hCMEC/D3 cell line extensively characterized for brain endothelial phenotype and used as a model of human blood–brain barrier function.

## Figures and Tables

**Figure 1 biomolecules-14-00390-f001:**
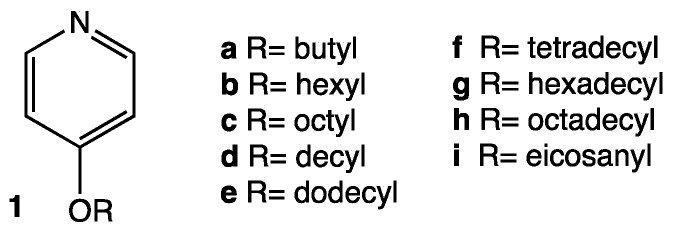
The structure of 4-alkoxypyridines **1a–i**.

**Figure 2 biomolecules-14-00390-f002:**
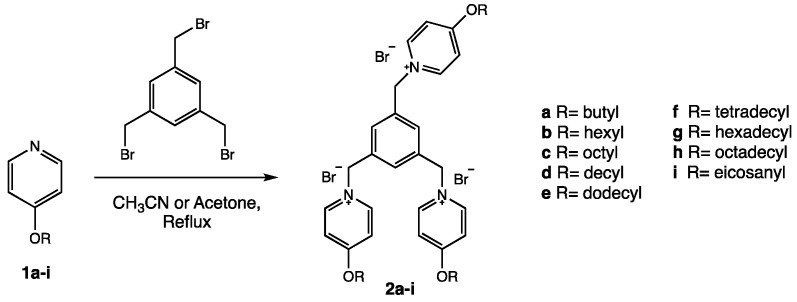
Synthesis of TRIFAPY compounds **2a–i** from alkoxypyridines **1a–i**.

**Figure 3 biomolecules-14-00390-f003:**
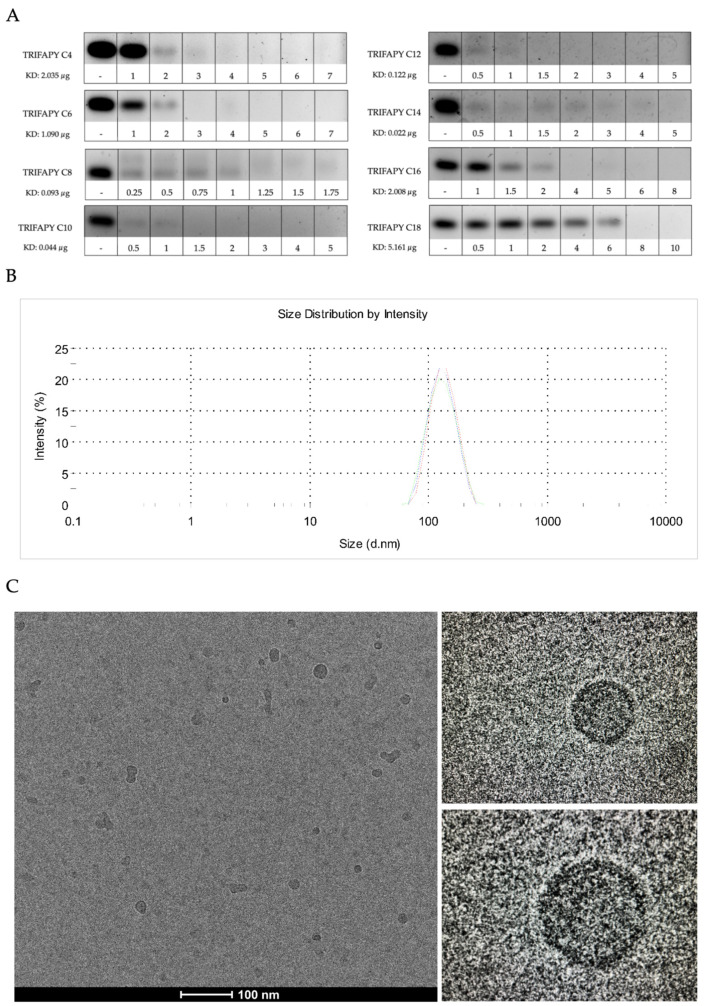
(**A**) Gel retardation assays. The binding capacity of the cationic agents was analyzed using 100 ng of FAM-HpScr9 incubated with increasing amounts of each transfectant expressed in µg. Reaction mixtures were incubated for 20 min and then resolved by electrophoresis in 0.8% agarose gels. Resulting bands were visualized by UV light in a Bio-Rad Gel Doc EZ™ Imager. Each assay was performed in triplicate, and a representative image of the bindings with each TRIFAPY is shown. The K_D_ value (expressed in µg/µL) was determined through nonlinear regression analysis using the One Site-specific Binding model in PRISM software. (**B**) Size of the complex PPRH-TRIFAPY C12 determined by DLS. Analyses were performed in triplicate (shown in blue, green and red), using 100 nM of PPRH and 2 µg/mL of TRIFAPY C12. (**C**). Cryo-TEM images. Captures of the complexes formed by TRIFAPY C12 and PPRH.

**Figure 4 biomolecules-14-00390-f004:**
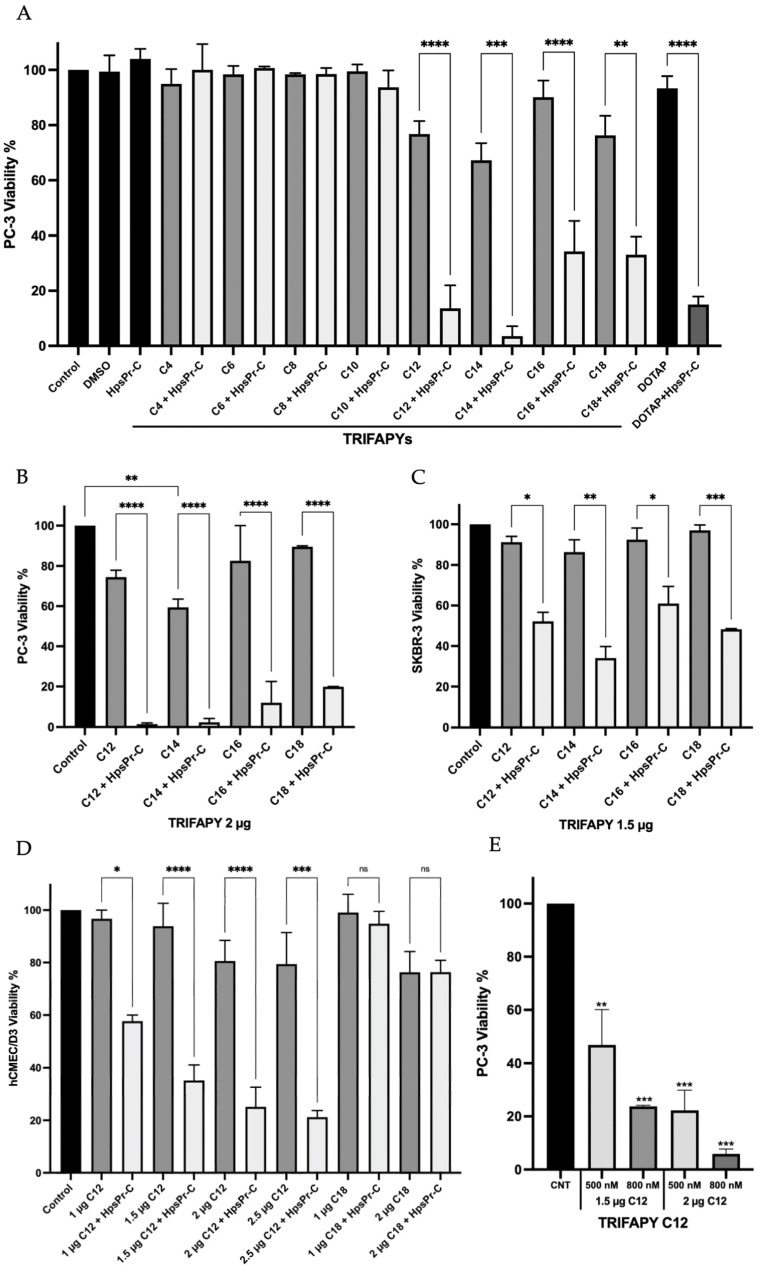
Cytotoxicity assays. (**A**) The intrinsic toxicity of each TRIFAPY was screened in PC-3 cell line at a concentration of 1.5 µg/mL. Subsequently, the effect caused by the transfection of 100 nM HpsPr-C in combination with the corresponding TRIFAPY or with 10 µM of DOTAP was assessed. (**B**) Selected TRIFAPYs C12, C14, C16, and C18 were tested in PC-3 cells at a higher concentration of 2 µg/mL to establish a range of working concentrations. (**C**) The intrinsic toxicity and transfection efficiency of the chosen TRIFAPYs were evaluated in SKBR-3 cells. (**D**) The inherent toxicity and transfection efficacy of TRIFAPY C12 were evaluated in the hCMEC/D3 blood–barrier cell line, with TRIFAPY C18 employed for comparative purposes. (**E**) Cell viability was determined upon treatment of PC-3 with the antisense oligonucleotide directed against survivin (aODN-Survivin) at two different concentrations 500 nM and 800 nM transfected using TRIFAPY C12 at 1.5 and 2 µg/mL. Statistical significance was determined using a one-way ANOVA with a multiple comparison test: (ns, non-significant; * *p* < 0.1, ** *p* < 0.01, *** *p* < 0.001, **** *p* < 0.0001). Experiments were performed in triplicate.

**Figure 5 biomolecules-14-00390-f005:**
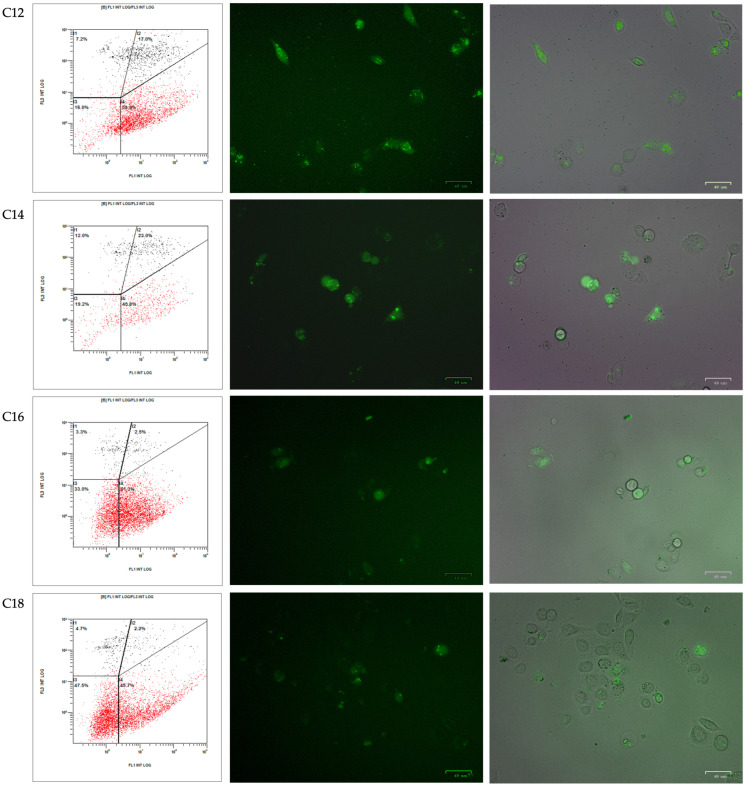
TRIFAPY-mediated internalization by fluorescence microscopy in PC-3. Images were captured 24 h after the incubation of the PC-3 cancer cell line with 100 nM FAM-HpScr9, in combination with 1.5 µg/mL of the corresponding TRIFAPY (**right panels**). Flow cytometry screenings were conducted in the same samples to assess the uptake of the labeled scramble PPRH in these cells (**left panels**). Lower quadrants represented the fluorescence emitted by viable cells (shown in red dots), while upper quadrants showed non-viable cells labeled with propidium iodide (shown in black). Right quadrants displayed the percentage of positive gated cells that internalized the fluorescent complex.

**Figure 6 biomolecules-14-00390-f006:**
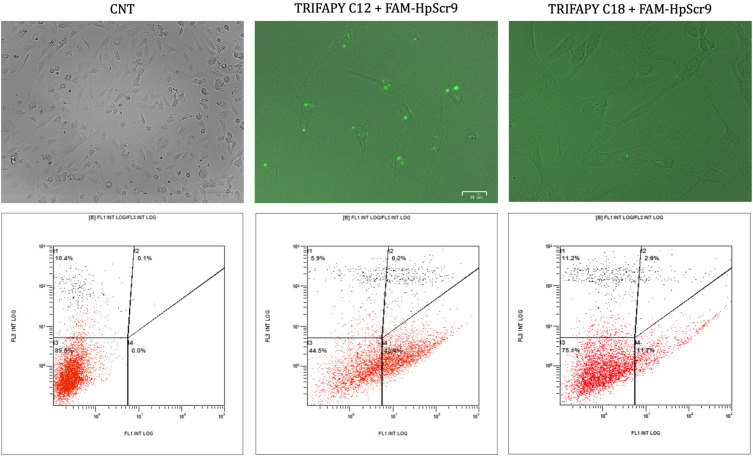
Cellular uptake determined by fluorescence microscopy and flow cytometry in hCMEC/D3 cells. Uptake experiments were performed after 24 h incubations of 100 nM of FAM-HpsScr9 with 1.5 µg/mL of TRIFAPYs C12 and C18. Interpretations of panels from flow cytometry and microscopy were as detailed in Figure 5.

**Figure 7 biomolecules-14-00390-f007:**
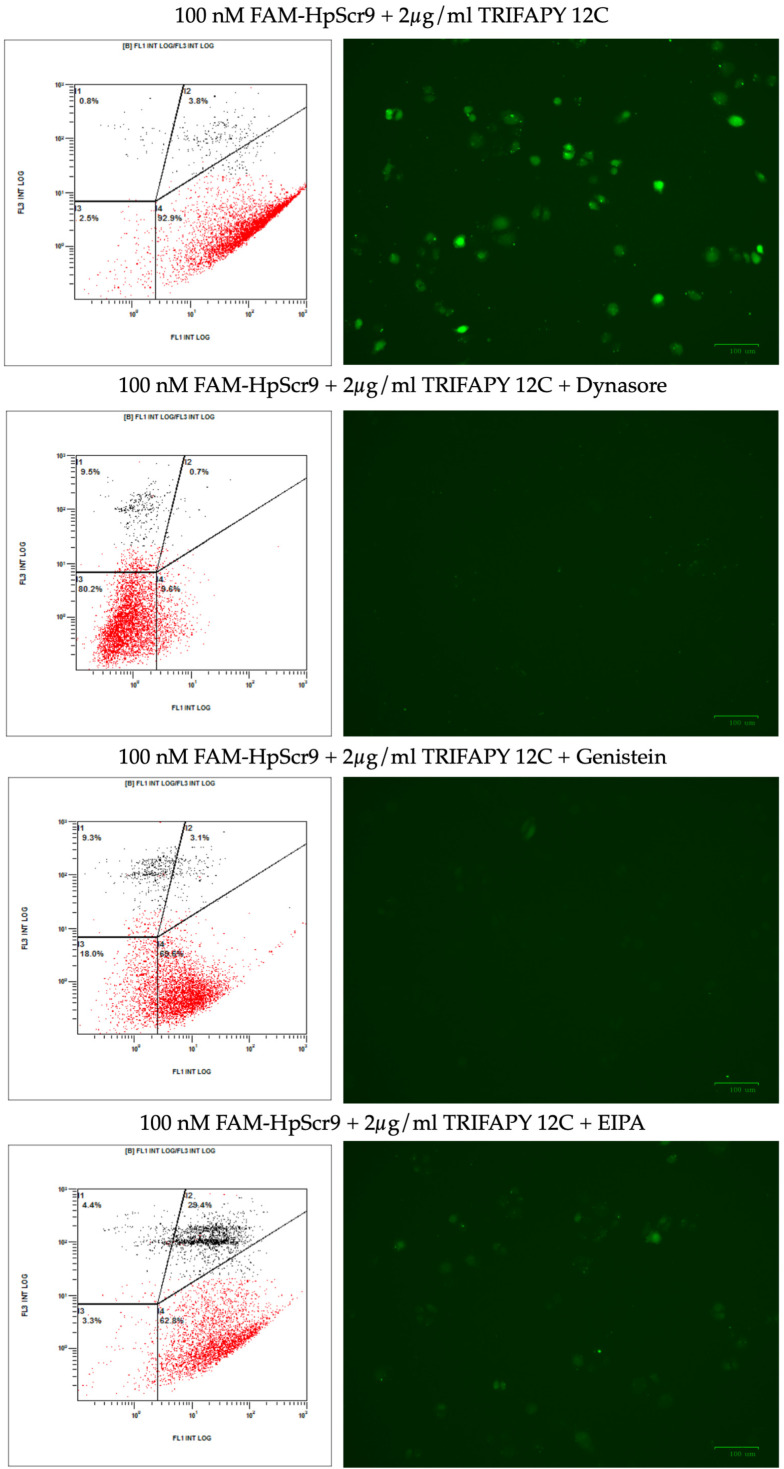
Endocytic pathways involved in complex internalization studied by flow cytometry and fluorescence microscopy. The entrance of the complex formed by PPRH-TRIFAPY C12 was determined in the PC-3 cell line in the presence of the endocytic inhibitors Dynasore (75 µM), Genistein (185 µM) and EIPA (33 µM). Incubation with inhibitors lasted for 1 h. Subsequently, cells were transfected with 100 nM of FAM-HpScr9 and 2 µg/mL of TRIFAPY C12. After 3.5 h of incubation, the percentage of fluorescence cells and X-MEAN parameters were determined by flow cytometry. The right panels show captures of fluorescence microscopy of the cells before proceeding to flow cytometry analyses.

**Figure 8 biomolecules-14-00390-f008:**
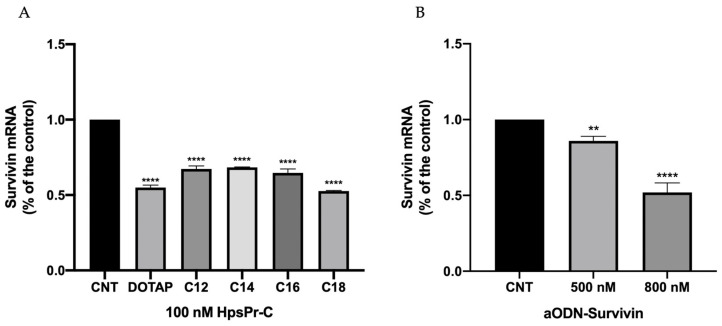
Survivin mRNA levels determined by RT-qPCR. (**A**) mRNA levels were determined in PC-3 cells transfected with 100 nM of HpsPr-C using 10 µM of DOTAP or 2 µg/mL of TRIFAPYs C12, C14, C16 and C18 for 24 h. (**B**) mRNA levels determined in PC-3 after 24 h treatment of 500 nM and 800 nM aODN-Survivin transfected using TRIFAPY C12 at 2 µg/mL. Statistical significance was determined through one-way ANOVA with a multiple comparison test (** *p* < 0.01; **** *p* < 0.0001).

**Figure 9 biomolecules-14-00390-f009:**
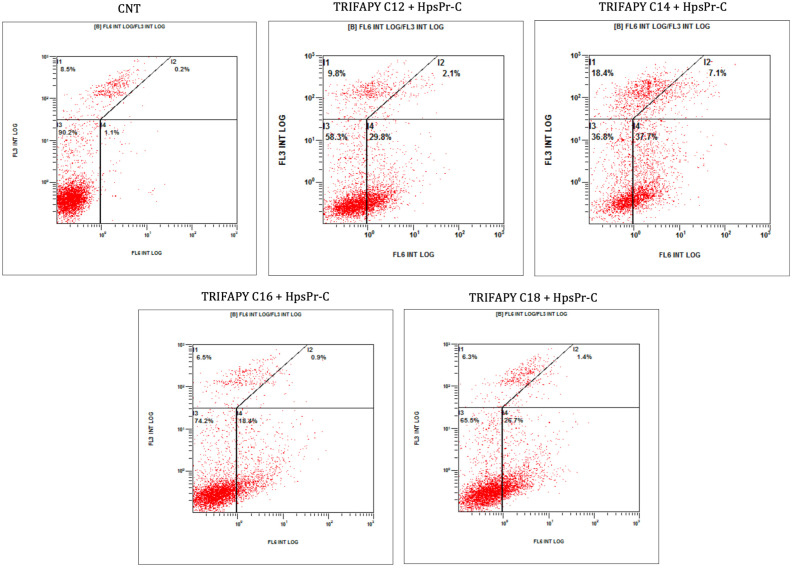
TRIFAPY-mediated apoptosis by flow cytometry. Apoptosis levels were determined by flow cytometry in PC-3 cells incubated with Annexin V, 48 h after the transfection with 100 nM HpsPr-C and 2 µg/mL of each TRIFAPY. Flow cytometry analyses divided the cell population into four quadrants. Both upper quadrants correspond to propidium iodide labeling, indicating cell death, whereas the lower left quadrant displayed the intrinsic autofluorescence of the cells. The upper and lower right quadrants showed the cell population labeled with Annexin V.

**Table 1 biomolecules-14-00390-t001:** Design of PPRHs and antisense oligonucleotide.

Gene	Oligonucleotide	Sequence 5’-3’	Location
Survivin	HpsPr-C	AGGGGAGGGATGGAGTGCAG T T T AGGGGAGGGATGGAGTGCAG T T	Promoter
Survivin	FAM-HpsPr-C	[6FAM]AGGGGAGGGATGGAGTGCAG T T T AGGGGAGGGATGGAGTGCAG T T	Promoter
-	Scrambled FAM-HpScr9	[6FAM] AAGAAGAAGAAGAGAAGAA T T T AAGAAGAAGAAGAGAAGAA T T	-
Survivin	aODN-Surv	GGGCAACGTCGGGGCACCCAT	Translation initiation

PPRH and antisense oligonucleotide sequences.

**Table 2 biomolecules-14-00390-t002:** Cellular uptake determined by flow cytometry in PC-3.

Treatment	% Positive Cells	X-MEAN
CNT	0.2	3.17
DOTAP + FAM-HpScr-9	69.0	64.4
TRIFAPY C12 + FAM-HpScr-9	76.9	31.9
TRIFAPY C14 + FAM-HpScr-9	68.8	31.4
TRIFAPY C16 + FAM-HpScr-9	63.8	11.9
TRIFAPY C18 + FAM-HpScr-9	47.9	30.7

Percentage of the internalization positive gated cells and X-MEAN in PC-3 cells treated with 1.5 µg/mL of each TRIFAPY or 10 µM DOTAP used for the transfection of 100 nM HpScr9-FAM.

**Table 3 biomolecules-14-00390-t003:** Cellular uptake determined by flow cytometry in hCMEC/D3 cells.

Treatment	% Positive Cells	X-MEAN
CNT	0.1	0.38
TRIFAPY C12 + FAM-HpScr-9	51.6	30.8
TRIFAPY C18 + FAM-HpScr-9	14.5	65.3

Percentage of the internalization positive gated cells and X-MEAN, 24 h after the transfection of hCMEC/D3 cells with 1.5 µg/mL of each TRIFAPY and the labeled PPRH HpScr9-FAM at 100 nM.

**Table 4 biomolecules-14-00390-t004:** Cellular uptake determined by flow cytometry after incubation with endocytic inhibitors.

Treatment	Inhibitor	% Positive Cells	X-MEAN
CNT	-	0.21	7.05
TRIFAPY C12	-	0.31	4.38
TRIFAPY C12 + FAM-HpScr-9	-	92.91	153
TRIFAPY C12 + FAM-HpScr-9	Dynasore	9.61	5.56
TRIFAPY C12 + FAM-HpScr-9	Genistein	69.61	12.8
TRIFAPY C12 + FAM-HpScr-9	EIPA	62.83	45.2

Percentage of positive gated cells that internalized the fluorescent PPRH transfected with TRIFAPY C12 upon inhibition of the endocytosis pathways with Dynasore, Genistein, and EIPA. It was also determined the X-MEAN parameter corresponding to the fluorescence intensity.

**Table 5 biomolecules-14-00390-t005:** Apoptosis levels determined by flow cytometry.

Treatment	% Apoptotic Cells
CNT	1.3
DOTAP + HpsPr-C	54.1
TRIFAPY C12 + HpsPr-C	31.9
TRIFAPY C14 + HpsPr-C	44.8
TRIFAPY C16 + HpsPr-C	19.3
TRIFAPY C18 + HpsPr-C	28.1

Percentage of PC-3 apoptotic cells was assessed 48 h upon transfection with each TRIFAPY at 2 µg/mL or 10 µM of DOTAP and the specific PPRH at 100 nM.

## Data Availability

All data are presented within the submitted manuscript.

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
