# Peer review of "Synthesis and Validation of TRIFAPYs as a Family of Transfection Agents for Therapeutic Oligonucleotides"

_biomolecules, 2024, doi:10.3390/biom14040390_

Round 1

Reviewer 1 Report

Comments and Suggestions for Authors

The paper by Isanta et al. describes an original family of prospective transfection agents
for cellular delivery of a specific polypurine hairpin type of triplex-forming oligonucleotide,
which has been previously studied as a potential gene silencing tool. The transfection agents
based on a symmetrical tripodand 4-alkoxypyridinium structure with long pendant alkyl chains
were employed to successfully deliver the oligonucleotidic hairpins into a row of cancer cell
lines, and decrease the level of anti-apoptotic target (survivin) mRNA.
I think this work may be an important contribution to the field of therapeutic
oligonucleotides, and I will be glad to see it published in Biomolecules after a few corrections,
which are specified below.
1. The main problem is a very peculiar structure of the survivin oligonucleotide to be delivered.
Its polypurine (dA and dG only) sequence is very specific and somewhat distant from either a
typical antisense oligonucleotide or siRNA. It would have been great if the authors could
additionally provide at least a few examples of how their liposomal system work with regular
mixed sequence oligonucleotides around 20some nt in length characteristic of many antisense
agents, and short RNA duplexes such as siRNAs. Inclusion of any examples of successful
delivery of the above, especially siRNAs, will tremendously expand the scope of the present
work and make it much more compelling.
2. Table 1: The structures of the hairpin oligonucleotides in the 2nd and 3rd rows appear to be
scrambled, which makes poor impression; may be better to present these in a pictorial format
rather than simple textual.
3. Unfortunately, the authors did not no compare the efficiency of their system with any other
common transfection agents such as PEI, which was used in ref 27.
4. There are no ‘oligonucleotide only’ controls without liposomes but with DMSO; there is also
no separate ‘DMSO only’ control.
5. Figure 1: replace ‘alcoxy’ with ‘alkoxy’.
Thus, it is formally a major revision as there are some essential controls missing.

Comments on the Quality of English Language

Only minor English editing is suggested.

Reviewer 2 Report

Comments and Suggestions for Authors

Authors describe synthesis of series of 1,3,5-tris[(4-alkyloxy-1pyridinio)methyl]benzene tribromides (TRIFAPYs) as potential delivery agents for therapeutic oligonucleotides. This is a valuable attempt to expand the range of currently available transfection agents, the chemical part is well-designed and executed, and the obtained compounds are sufficiently characterized. However,  before accepting this work for publication, some issues should be clarified. One of the most confusing is the structure (and nomenclature) of adducts formed by TRIFAPYs and the oligonucleotides. Authors use names such as liposomes, complexes, DNA-liposome complex, or nanoparticles - these terms do not mean the same thing

At the same time, the authors nowhere provide evidence that the particles they obtain are liposomes. This is necessary for a better understanding of the way in which TRIFAPYs participate in oligonucleotide transport through cellular membrane.  Liposomes are assemblies with a precisely defined structure of concentric vesicles in which a membranous lipid bilayer surrounds an aqueous volume. Are the oligonucleotides enclosed inside the liposome (if it is formed), or are they adsorbed on its surface? Or do the TRIFAPYs rather coat the oligonucleotides as in the case of e.g. lipofectamine, which is suggested by the term “complex”?

Imaging of the structures formed by TRIFAPYs by transmission electron microscopy should be a first step. DLS measurement of “the complex PPRH-TRIFAPY C12”  does not determine whether the observed particles are liposomes or simply aggregates of the TRIFAPYs complexed with oligonucleotides.

Please also briefly discuss the observation that "each transfection agent had a different ability to bind DNA from PPRH depending on the length of its carbon chain". The fact that small differences in the length of the carbon chain, especially in the case of longer chains such as C12, C14, C16, C18, have such an impact, is not obvious.

Finally, all studies performed for TRIFAPYs (cytotoxicity, internalization, mechanism of entrance, silencing activity)  should be conducted with lipofectamine or another, commonly used, transfecting agent for comparison.

Comments on the Quality of English Language

Manuscripts generally benefit from careful re-reading before submitting the revised version. I also recommend running a spell check.

Round 2

Reviewer 1 Report

Comments and Suggestions for Authors

The authors have introduced all the changes suggested by this Reviewer.

It is recommended to publish the current version as is.

Comments on the Quality of English Language

English is fine. Apart from usual editorial spellcheck, no issues were detected.

Reviewer 2 Report

Comments and Suggestions for Authors

All my concerns have been addressed. I recommend the publication of this work.

PS Please provide a scale on the enlarged cryo-TEM images of the complexes formed by TRIFAPY C12 and PPRH in Figure 3.

Please note a discrepancy between the statement “The complexes had a size of 125 nm as determined by DLS, forming well-defined concentrical vesicles as visualized by Cryo-TEM” and the statement “….we conducted cryo-TEM analysis, and the results demonstrated the absence of aggregates but well-defined concentrical vesicles of less than 50 nm”.  

The more globular the shape of a molecule is (and the TRIFAPY C12 and PPRH complexes are “well-defined concentrical vesicles”,  the better the Dh value should reflect the actual diameter of the observed in cryo-TEM particles.